# Usability, User Experience, and Acceptance Evaluation of CAPACITY: A Technological Ecosystem for Remote Follow-Up of Frailty

**DOI:** 10.3390/s21196458

**Published:** 2021-09-27

**Authors:** Rodrigo Pérez-Rodríguez, Elena Villalba-Mora, Myriam Valdés-Aragonés, Xavier Ferre, Cristian Moral, Marta Mas-Romero, Pedro Abizanda-Soler, Leocadio Rodríguez-Mañas

**Affiliations:** 1Biomedical Research Foundation, Getafe University Hospital, 28905 Getafe, Spain; 2Centre for Biomedical Technology (CTB), Universidad Politécnica de Madrid (UPM), Pozuelo de Alarcón, 28223 Madrid, Spain; elena.villalba@ctb.upm.es (E.V.-M.); myriam.valdes@salud.madrid.org (M.V.-A.); xavier.ferre@ctb.upm.es (X.F.); cristian.moral@ctb.upm.es (C.M.); leocadio.rodriguez@salud.madrid.org (L.R.-M.); 3CIBER of Frailty and Healthy Aging (CIBERFES), 28001 Madrid, Spain; pabizanda@sescam.jccm.es; 4Centro de Investigación Biomédica en Red en Bioingeniería, Biomateriales y Nanomedicina (CIBER-BBN), 28029 Madrid, Spain; 5Geriatrics Service, Getafe University Hospital, 28095 Getafe, Spain; 6Geriatrics Service, Albacete University Hospital, 02006 Albacete, Spain; mmasr@sescam.jccm.es; 7Faculty of Medicine, University of Castilla-La Mancha, 02008 Albacete, Spain

**Keywords:** frailty, home monitoring, user-centered design, usability, user experience, acceptance

## Abstract

Frailty predisposes older persons to adverse events, and information and communication technologies can play a crucial role to prevent them. CAPACITY provides a means to remotely monitor variables with high predictive power for adverse events, enabling preventative personalized early interventions. This study aims at evaluating the usability, user experience, and acceptance of a novel mobile system to prevent disability. Usability was assessed using the system usability scale (SUS); user experience using the user experience questionnaire (UEQ); and acceptance with the technology acceptance model (TAM) and a customized quantitative questionnaire. Data were collected at baseline (recruitment), and after three and six months of use. Forty-six participants used CAPACITY for six months; nine dropped out, leaving a final sample of 37 subjects. SUS reached a maximum averaged value of 83.68 after six months of use; no statistically significant values have been found to demonstrate that usability improves with use, probably because of a ceiling effect. UEQ, obtained averages scores higher or very close to 2 in all categories. TAM reached a maximum of 51.54 points, showing an improvement trend. Results indicate the success of the participatory methodology, and support user centered design as a key methodology to design technologies for frail older persons. Involving potential end users and giving them voice during the design stage maximizes usability and acceptance.

## 1. Introduction

### 1.1. Research Context

Intrinsic capacity, according to the World Health Organization (WHO), is the combination of the physical and mental (including psychological) capacities of the individuals. Intrinsic capacity is thus part of functional ability together with the environment and the interactions with it. The concept of frailty is closely related and complementary to intrinsic capacity. Frailty can be defined as a stage of age-related decline reducing intrinsic capacity and functional reserve of older persons, thus predisposing them to adverse events (mortality and disability, among others). These days, there is a pressing need to develop comprehensive community-based approaches and to introduce interventions to prevent functional decline [1]. 

The risk of developing chronic conditions, including disability and dependency, increases with age [2,3], and is changing the classical approach to manage functionally declining older persons. Considering that functional decline is accompanied by a loss in functional reserve, it is very unlikely that disability is reversed. In this way, healthcare systems need to move towards person-centered approaches that anticipate the earliest stages of functional decline (i.e., frailty) to prevent disability, since becoming frail can be delayed, slowed, or even reversed.

Estimated prevalence of frailty is 18% (95% CI: 15–21%), and it seems to be correlated with age, gender (female), and socio-economic factors such as lower education and wealth [4]. Good news is that frailty is reversible, but to achieve it, it is of paramount importance to fight inactivity and sedentariness [5]. Scientific literature supports activity-centered interventions to delay and even reverse frailty and disability [6,7,8,9,10,11]. Furthermore, interventions on nutrition, such as modifying habits, increasing protein and micronutrient intake, are also recommended [12,13], as well as interventions on inadequate drug prescriptions [14,15,16]. And finally, it is also important that the physiological and social aspects are not left apart [11]. 

A frail older person usually shows decreased neurological and muscle function [17], normally accompanied by an accelerated involuntary weight loss and a decline in the skeletal muscle [18]. Moreover, considering the results published in a relatively recent systematic review, in the 30.6% of the studies that were analyzed, associations between gait speed, disability, frailty, sedentary lifestyle, falls, muscular weakness, diseases, body fat, cognitive impairment, mortality, stress, lower life satisfaction, lower quality of life, and poor performance in quantitative parameters of gait were found [19].

Ageing in Place purses that older persons continue living at their homes as they age [20], which brings along important economic benefits given the reduction of the institutionalized care [21]; information and communication technologies (ICTs) can play a crucial role to promote it [22]. For instance, having fresh and periodic information on variables associated to poor health outcomes (e.g., gait speed, muscle power, and involuntary weight loss) can be a great asset to trigger early interventions to prevent disability and dependency. Smart home technologies [23,24,25,26,27], wearable sensors [28,29] or mHealth technology [30] may enable continuous, ubiquitous and transparent monitoring of the independent older adult, supporting the traditional geriatric approach to identify older people at risk of disability. Notwithstanding, more effort is still needed to assess not only how reliable and valid the ICT-based approached to measure frailty are, but also deeply study the associated ethical, technical, and economic issues [31].

Nevertheless, the lack of consensus in terms of technology acceptance by older persons must be considered. Several authors have reached the conclusion that older persons are not interested in innovative technologies [32,33], while others state that older people have already accepted new technologies, mainly because they have been proven useful in meeting their information needs, especially in health [34]. Yet it seems that the use of ICTs by the older population is strongly linked to physical limitations (e.g., abilities, chronic illnesses, etc.), mental limitations (e.g., fear of damaging the technology, electric shocks, making mistakes, etc.), educational limitations (e.g., low levels of literacy, limited electronic literacy, learning barriers, etc.), structural limitations (e.g., design of the appliance), instructional limitations (e.g., instructions on how to use a technology are hard to follow) and to a limited access to the technology (e.g., financial costs) [35,36].

The design of those technologies to be used by older persons must be done according to their characteristics. Methodologies such as user-centered design (UCD) [37] and participatory design (PD) are a good alternative to develop right solutions for a specific audience [33,38], since they help designers better understand the environment of use. Older people are usually excluded from product design activities since they are stigmatized as people reluctant to engage with technology, and this is probably one of the primary causes that prevent older people becoming loyal users of technological solutions [38,39].

The current demographic challenge is forcing researchers to focus on discovering feasible alternative ways of providing healthcare to the older population who are at an increased risk of suffering adverse events [40]. And, as it has been mentioned earlier, ICTs may help identifying early risk indicators for adverse events, providing a means for self-managing them. To this extent, its use in the context of frailty still at the very beginning [41].

### 1.2. Objective

The main objective of this work is to evaluate the usability, user experience (UX), and acceptance of older persons’ interaction layer of CAPACITY, a frailty home-monitoring system aimed at the prevention of disability.

The manuscript is structured as follows. First, the CAPACITY ecosystem is presented as a modular infrastructure to monitor frailty and prevent disability. Second, the specific methodology followed in this work is described to later present and discuss the obtained results. Finally, conclusions are extracted and future work proposed.

## 2. Materials and Methods

### 2.1. Overview of the CAPACITY Technological Ecosystem

CAPACITY is a technological ecosystem aimed at preventing disability among the older population by detecting and intervening regarding frailty; it also provides a substrate to connect all relevant people in the care process (see Figure 1). Using CAPACITY, the older population can be remotely supervised by community care professionals. So, in case worrying declines are detected, specialists (i.e., geriatricians) can be included in the loop. Intervention provided to older persons is grounded on three main pillars: the VIVIFRAIL physical activity program (declared as a success story by the European Commission) [42], personalized nutritional recommendations, and a program to detect risk of polypharmacy.

A Randomized Clinical Trial (RCT; ClinicalTrials.gov Identifier: NCT03707145) has allowed demonstrating that CAPACITY is an effective tool for a fast improvement of the frailty status as well as to reduce the use of healthcare resources [41].

### 2.2. CAPACITY Interaction System

CAPACITY services are offered through a set of user-adapted mobile applications. The functionalities offered to older persons are:Unceasing intrinsic capacity follow-up that enables triggering potential deterioration alarms;Access to a customized therapeutic plan (intervention), given the peculiarities and needs of the older person;Retrieving their own evolution;Communication with formal carers via asynchronous channels; andNotifications on pertinent alarms related to health.

Apart from helping older persons, CAPACITY also offers different functionalities to other relevant stakeholders, namely primary and specialized care professionals, and informal caregivers, as shown in Figure 2. Work published [43] contains a wider description of all functionalities and services offered by the CAPACITY technological ecosystem to all involved people. In any case, this work solely focuses on the older persons as they are the center of the care.

Older persons being followed by CAPACITY need to use a home monitoring kit aimed at measuring variables with high predictive value for adverse events. This monitoring system consists in a gait-speed sensor [44], a sensor to indirectly (through the chair stand test) measure power in the lower limbs [45], and a wireless commercial weight scale to measure involuntary weight loss. Figure 3 illustrates the prototypes of the sensors originally designed for CAPACITY. 

The interaction with the home monitoring kit is handled by a mobile application that acts as a guiding element to the older person, as a data concentrator (bluetooth connection with the monitoring kit; see Figure 3 and Appendix A for details), and as data input point, not only enabling the older adult using the sensors but also completing a set of questionnaires to enrich the information handled by the clinical professionals. These questionnaires are adapted versions of the Frailty Phenotype criteria [46], Mini Nutritional Assessment (MNA) [47], Barthel Index [48], FRAIL Scale [49], and the Functional Activities Questionnaire (FAQ) [50]. 

This interaction system was iteratively designed under a user-centered approach. Different prototypes were created and tested, first in a laboratory environment, later in a clinical environment, and, finally, at the final users’ dwellings. In the two last cases, the system was evaluated by older users. The outcomes of each iteration allowed designers improve and adapt the interaction system to the needs, preferences, and context of use of the older adults. 

Figure 4 shows how the interaction system evolved during the process. An in-depth description of this iterative process and the resulting interaction system can be found in [51].

Figure 5 shows the workflow that needs to be followed to interact with any of the components of the monitoring kit. The process starts with the app notifying a pending measurement (prescribed by the physician as part of a personalized follow-up) and the user pushing the corresponding button to start it. Then, the older person is provided with a short explanatory video showing how the measurement will be performed. Once the user is ready, that is explicated by pressing a specific button, the actual measurement takes place; this part is fully guided by voice commands and accompanying pictures (e.g., ‘please switch the sensor on’, ‘please sit on a chair’, ‘the process will start after the countdown’, etc.). Transparently to the user, the app and the sensor establish a bluetooth communication channel used to register the datum. Once the process is over, the older person receives a confirmation with some feedback related to the measurement.

### 2.3. Assessment Tools

The usability, UX, and user acceptance related to the CAPACITY technological ecosystem has been assessed. 

Usability, defined as the extent to which a product can be used by specified users to achieve specified goals with effectiveness, efficiency, and satisfaction in a specified context of use [52], has been assessed using the system usability scale [53,54]. SUS is a short 10-item Likert questionnaire that provides a measure of people’s subjective perceptions of the usability of a system; concretely, SUS focuses on learnability and usability, which are indeed correlated [55]. These 10 items can be evaluated from ‘1—fully disagree’ to ‘5—fully-agree’. The total score ranges from 0 to 100. Although SUS is a simple tool, a study carried out by Tullis [56], who compared the effectiveness and accuracy of five questionnaires for assessing usability across different sample sizes, reached the conclusion that it is a reliable scale, especially when the sample is over 12 users.

UX has been assessed with the user experience questionnaire [57]. UEQ does not provide an overall score, but a score related to six different categories: attractiveness, perspicuity, efficiency, dependability, stimulation, and novelty. The score for each category is calculated by averaging the different items within it; each item’s value ranges from −3 to 3, where extreme values represent two opposite concepts (e.g., attractive vs. unattractive). UEQ was included as an evaluation tool to complement the domains’ SUS addresses.

Acceptance has been evaluated using the technology acceptance model [58] (adapted to the use case. See Appendix A for details) and a customized short quantitative questionnaire. TAM evaluates, throughout 12 items (answers range from ‘1—fully disagree’ to ‘5—fully-agree’), two different categories: perceived usefulness and perceived ease of use. The maximum score is 60 (30 points in each category); final scores are calculated by averaging. To further investigate acceptance, a customized acceptance interview consisting in a Likert-type scale (from 1 to 5, the same as SUS) assessing all three components of the home monitoring kit was used. This interview had the following structure:Q1: The information the device provides motivates me to have a healthier lifestyle;Q2: The device makes me feel cared for;Q3: Using the device is a burden for me;Q4: The device enables me to control my own health; andQ5: I would use it.

For all the scales and questionnaires, Spanish versions were used (local language).

### 2.4. Recruitment and Data Collection

The impact of administering a multicomponent intervention partially supported by the CAPACITY ecosystem was assessed by conducting a pilot, prospective, randomized, and blind study. The pilot study lasted 12 months: 6 months were dedicated to recruitment, and 6 months to intervention. Within this wider experiment, where the primary endpoint was to investigate whether the proposed technology helped preventing or reversing frailty, usability, UX, and user acceptance were evaluated as secondary endpoints (along with others). The pilot study was carried out simultaneously in two institutions: Getafe University Hospital and Albacete University Hospital. 

Participation criteria were:Inclusion criteria:o+70 years old;oLiving at home;oBarthel index [48] ≥90; andoBeing pre-frail or frail.


Exclusion criteria:
oInadequate home infrastructure impeding the installation of the technology;oInability to understand how to use CAPACITY;oMedical condition incompatible with the VIVIFRAIL physical activity program;oHistory of drug/alcohol abuse;oPsychiatric disorders;oLiving with a participant; andoParti9cipating in another interventional clinical study.


Pre-frail participants were those meeting two Frailty Phenotype criteria [46] and suffering from at least four comorbidities, since they are the ones with the highest risk for developing frailty. Frail individuals were those meeting at least three Frailty Phenotype criteria and having at least four comorbidities.

Two research groups (arms) were defined. A control group receiving usual geriatric care and an intervention group who received the same multicomponent intervention but partially supported by the CAPACITY system. Stratified randomization by age (70–85, +85), sex (male, female), diagnosis (pre-frail, frail) and educational level (non-formal education, higher education, others) was applied to ensure groups were balanced.

Sample size could not be empirically calculated due to a lack of similar studies aiming to the same primary endpoint, so it was established to 90. Reasons behind this decision were:There were two different groups of interest (i.e., pre-frail and frail older persons);Given the usual standards, a recruitment objective of 20 subjects per interest group and research arm was set, for a total of 80; andResearchers assumed that a potential dropout rate of 10–15% over the previous calculation, so the target sample size was increased to 90.

Data reported in this manuscript are restricted to those participants who were randomly allocated into the intervention group (*n* = 46), since they were the only ones that used the CAPACITY system during the six months of intervention. The modules that supported the intervention were: (1) monitoring system, (2) evolution of the older person (e.g., access to follow-up information collected by the home-monitoring kit), and (3) basic asynchronous communication. All technological components were preconfigured prior to the delivery to the participating older persons (i.e., a tablet was delivered with the app already installed and configured to receive data from the home monitoring kit), so they only had to follow notifications and instructions. Besides, older participants received an initial training during the installation of the technology in their homes. This face-to-face training was delivered once and lasted approximately one hour. During the session, a user manual was provided that was used as a reference to show all functionalities to the older person, who had to repeat what was learnt (e.g., how to measure gait speed or complete a questionnaire). After this session, a telephone line remained open during the weekdays at working hours to attend any consultation or issue coming from the older participants.

Data related to usability and acceptance were collected at baseline and after three and six months of intervention. SUS and TAM were registered in all three sampling points while UEQ and the ad hoc acceptance questionnaires were only administered in the last data collection point to enrich the collected data with UX information and prospective acceptance.

## 3. Results

A total of 46 older persons used the CAPACITY technological solution to undergo an intervention aimed at preventing/reversing frailty; 14 were male (30.43%) and 32 female; mean age was 82.11 (SD = 5.42) years old. Regarding educational level, 20 of participants using technology did not have formal education (43.48%), 20 had primary studies (43.48%), 5 received secondary education (10.87%), and 1 received higher education (2.17%). Finally, most of the participants (30 persons -65.22%-) did not have any previous experience with technology (i.e., smart phones and internet) while 9 of them (19.56%) used it in a daily basis; the remaining part made an occasional use of the technology (3 subjects -6.52%-) or had used it once or twice before this study (4 subjects -8.70%-). Table 1 shows the description of the population that participated in the study.

Nine participants dropped out from the study during the follow-up period, raising a final sample for analysis of 37 subjects. All subjects completed the questionnaires about usability, UX, and acceptance evaluation at the second visit (three months) and at the end of the follow-up; 25 participants completed it at baseline.

Table 2 depicts the adherence to the monitoring plan calculated as the average commitment to the measurements that the users needed to perform as part of their treatment. Full adherence (100%) means that all participants performed all prescribed measurements. Table 2 also shows the default periodicity for the different measurements, but it must be taken into consideration that additional measurements could be requested. For all measurements a push notification was sent to the user through the app.

Table 3 shows the usability results. Usability obtained averaged SUS values of 80.11/100 (SD = 13.66) at baseline, 83.31/100 (SD = 15.07) at month 3 and 83.68/100 (SD = 1.62) at the end of the study.

Table 4 depicts the results regarding UX, that was only assessed at the end of the intervention. Averaged values of 2.20/3 (SD = 0.64) in terms of attractiveness, 2.30/3 (SD = 0.73) in perspicuity, 1.99/3 (SD = 0.75) in efficiency, 2.16/3 (SD = 0.66) in dependability, 2.05/3 (SD = 0.72) in stimulation, and, finally, 2.09/3 (SD = 0.98) in novelty were obtained.

Table 5 contains those results corresponding to assessing the acceptance of the CAPACITY solution in terms of TAM, that show an improving trend (*p* = 0.15) starting with a value of 49.00/60 (SD = 8.24) at baseline, that gets to 50.68/60 (SD = 6.68) at month 3 and reaches 51.54 (SD = 6.97) at month 6. On the other hand, Table 6 presents the results related to the administration of the ad-hoc quantitative questionnaires, that evaluate individually each component of the home-monitoring kit.

SUS and TAM data were analyzed according to the educational level (i.e., non-formal education, primary education, secondary education, or higher education), living conditions (i.e., alone, with younger relatives, or with other older person), daily help received (i.e., from nobody, from a younger relative, from other older person, or from social services), previous experience with technology (i.e., no experience, used it once or twice before, occasionally used, or daily use), and frailty diagnosis (i.e., pre-frail, or frail). Table 7 shows the evolution of the reported SUS and TAM according to the category labels. 

Statistically significance related to the evolution= of the reported SUS and TAM within the categories described above has been analyzed. Only those older persons living with a younger relative showed a marginal but significant improvement in the reported SUS between baseline and month 3 (*p* = 0.049).

## 4. Discussion

This research study shows that the CAPACITY technological ecosystem has a very high-performance in term of usability, UX, and acceptance. Results have been obtained in a real-world scenario, where pre-frail and frail older persons used CAPACITY as the main vehicle to avoid transiting to disability.

Usage information demonstrates a high adoption rate, with an average adherence to the monitoring plan very close to or matching 100% for all components of the monitoring plan (i.e., use of sensors and completion of questionnaires). This endorses the validity of data collected in terms of usability, UX, and acceptance, since these are based on an intensive use of the system under assessment. However, the high usage of the system is not fully correlated with the expected use; for instance, during the experimentation the physicians detected that some users were not complying with the temporality of the monitoring plan (i.e., some measurements were missing, and they had to reach out to the older person to remind him or her). This has a twofold interpretation: on the one hand, sometimes notifications are neglected by the users, implying that new strategies should be found to promote prompt responses; but, on the other hand, the information that is constantly being provided to the clinical team allows a closer follow-up of the older persons, enabling early actuation of potentially worrying situations.

Based on the data collected by Sauro [59], the mean SUS score across a large number of usability studies is 68. If that value is used as a reference, the mean SUS obtained in all sampling points is highly above average. Furthermore, according to the Sauro–Lewis SUS grading curve [60], obtained score would be qualified as an A, with the last measurement really close of reaching A+, set at 84.1. So we can state that the evaluated user interaction is perceived as very good, almost excellent [61]. However, although usability seems to improve with use, obtained paired data Student’s test does not demonstrate that this improvement is statistically significant; a plausible explanation for this non-significant result could be linked to a ceiling effect, probably associated to an insufficient sample size. A further analysis by category showed that those older users living with a younger person marginally but significantly improved reported SUS between baseline and month 3 (*p* = 0.049), but this isolated result does not entitle to draw any solid conclusion since no other significant differences were observed.

Although usability is very high, which implies the UCD process was highly successful, it is not the highest possible, so there is still room for improvement. Most of the averaged items scored very close to the edges of the scale, which is good for the evaluation of the system, but some others deviate a bit from the expected value and are those which are susceptible to be improved. SUS items Q4, Q7, and Q10 are the ones lowering the overall score (without significant changes along the follow-up, except for Q4, that seems to show an improvement trend). The description of these items is:Q4: I think that I would need the support of a technical person to be able to use this system;Q7: I would imagine that most people would learn to use this system very quickly; andQ10: I needed to learn a lot of things before I could get going with this system.

These relatively low evaluations in these items could be linked to the unfamiliarity or insecurity of the older adults who used the technology during the intervention (65.22% of the sample did not have any experience with technology). Actually, obtained results indicate that, after six months of use, there are statistically significant differences in the reported SUS depending on the previous experience with technology (*p* = 0.017), which implies that its relationship with reported SUS needs to be further investigated.

UEQ does not provide an overall score for the UX but an individual score for each category. Scores between −0.8 and 0.8 usually represent a neutral evaluation, while values over 0.8 represent a positive evaluation; values below −0.8 represent a negative evaluation [62]. Obtained results are exceptional since all categories received averaged scores higher to or really close to 2, and given the fact that extreme UEQ values are very uncommon [63]). Furthermore, lower bounds of all confidence intervals per category are significantly above the minimum threshold established to be considered as positive evaluations (*p* = 0.05).

UX results in terms of UEQ have been benchmarked using a dataset with data from 9905 responses corresponding to 246 studies [64]; however, and given the fact that product categories have been not considered, this benchmarking can only be used as a first indicator to assess the UX of the system under study. Figure 6 represents the result of the benchmarking; in all six categories CAPACITY ranks over average (i.e., top 10%).

It is important to analyze whether the UEQ respondents have provided random answers to endorse validity of the obtained results. In this case, given the specific characteristics of the population who participated in the study (i.e., older persons with poor education background and digital literacy), some inconsistencies (i.e., all items in a scale should measure a similar UX quality aspect; if there is a big difference—>3—this is an indicator for a problematic data pattern) in the provided answers have been found for several respondents. These inconsistent answers can be due to misunderstanding of one or several items. One respondent was inconsistent in three categories, six in two categories, and eight in one category. According to Schrepp [62], answers to UEQ with two or more inconsistencies should be considered suspicious. No significant changes are observed when the doubtful information is removed from the analysis: all six categories stay with averaged values above two; in the same way, all categories remain qualified as excellent in the benchmarking.

Acceptance in terms of TAM reached a maximum score of 51.54 in the last sampling point, also showing an increasing trend amongst data collection points (*p* = 0.15 from baseline to month 6); given the small sample size of this research work, and the fact that statistical significance is a function of both the sample size and the magnitude of the estimated effect, *p*-values lower than 0.2 could be considered statistically significant [65,66]. Furthermore, Student’s *t*-test was significant (*p* = 0.02) for the positive evolution of the perceived ease of use between baseline and month 6. All individual items obtained averaged values above 4 in the last sampling point. The item that took the longest in reaching a value of 4 was Q2, under the category of ‘perceived usefulness’; this item relates to whether users perceived that CAPACITY contributed to his or her daily life independency. A possible explanation could be related to the fact that results associated to a physical intervention are not perceived immediately. In any case, this seems to be more related to the clinical aspects of the project rather than to the technological ones. On the other hand, the evaluation of each individual device (i.e., each component of the home monitoring kit) indicates very high acceptance: all questions pursuing a value of 5 got an average value over 4, while those aiming at 1 got values below 1.5. Acceptance results not only suggest that the older population would accept using and having CAPACITY devices at home as a way of being constantly monitored in terms of function, but also that all components of the monitoring kit are perceived as empowerment tools to motivate having a healthier lifestyle and to control his or her own health.

Not many RCTs exist in the field of ICTs applied to frailty management, which prevents the availability of a wider number of works focused on assessing the usability of technologies for treating frail population [41]. Works analyzing usability-related aspects of technology during real interventions also report satisfactory results [67,68], however, since the design procedure followed is scarcely described, the sample size is significantly smaller than the one presented in this paper, and data related to adherence are not optimal (i.e., far from 100% adherence as reported in this paper), those results should be interpreted with caution. On the other hand, the majority of the research addressing how older persons interact with technology is done in controlled environments and under the supervision of domain experts [69,70,71,72,73]. Most of the published related research use standardized tools such as those used in this work, which is aligned with the methods followed in the current approach; moreover, despite the heterogeneous approaches in terms of the target application, ranging from rehabilitation [67,68,69] to exergames [70], monitoring cognitive impairment [71], fall risk [72], or evaluating available health apps [73], the used interaction instruments, including mobile devices [67,69,72,73], personal computers [68,71], or custom prototypes [70], and the diverse characteristics of the target population, that in some cases have a previous experience with the technology to be used [68] and in some others cannot use it without help [71], the UCD approach is a commonality backing almost all approaches from a methodological perspective.

## 5. Conclusions

The objective of this research work was to investigate the usability, UX, and acceptance of CAPACITY, a technological ecosystem to prevent disability. This objective has been achieved, obtaining very satisfactory results in all domains under study. The usability of CAPACITY (in terms of SUS) was rated as almost excellent, and UX (in terms of UEQ) as excellent; finally, the proposed technology, both from a software and a hardware perspective, seems to be highly accepted by the target population (in terms of TAM and ah-hoc questionnaires). Besides, adherence to using CAPACITY has been found optimal, which implies both that these superb results are correlated with maximizing the actual use of the proposed solution in a real environment and that data supporting the conclusions are based on reliable and solid information.

The main contribution of this paper is thus the demonstration that following an iterative UCD approach starting in a controlled laboratory environment to come up with a pre-validated interaction system, and later upscale it to a real uncontrolled environment is a valid strategy to maximize usability, UX, acceptance and actual adoption of a system. Furthermore, this research work contributes with a new experience to the scant number of RCTs studying how pre-frail and frail older persons interact with technology.

Findings support UCD as a key methodology. Involving potential end users and giving them voice during the design stage maximizes usability, UX, acceptance and usage. In this research, older persons were involved from the very beginning: first, older people’s opinions were captured in a laboratory environment to later move towards clinical and home environments. Insights collected during this process enabled obtaining these excellent data within a RCT. Results indicate a potential high adoption in a wider deployment scenario (i.e., production phase). Some limitations must be taken into consideration when interpreting the results presented in this manuscript. First, the relatively small sample implies that findings need to be construed with prudence. Second, the external validity of the findings is not clear (i.e., whether the tested interaction system would obtain equivalent result in a population with different characteristics, such as culture, education, experience with technology, etc.); moreover, the assessment tools used to measure usability, UX, and acceptance, although they have been conceived to provide objective measurements, are highly dependent on the subjectivity of the respondents, so what is really measured is a perception on the different explored domains, given that humans are prone to bias while rating their experiences after interacting with a system. Third, no data related to the use of the technical assistance telephone line available for participants were collected, which has prevented integrating that information in the interpretation of the results. And, finally, no information on patient–physician communication through the platform was registered, limiting the extent of the presented usage analysis.

The CAPACITY technological ecosystem is constantly being improved, and new services added. From a service perspective, the current version of the solution incorporates functionalities to support a novel organizational model that interconnects all relevant people in the care process: the older person, the informal caregiver, and the primary and specialized care professionals. This evolved version of CAPACITY also integrates mechanisms (algorithms) to automatically detect functional decline and alert professionals and means to provide a multicomponent. Future work includes carrying out a new multicentric field experimentation (RCTs in Spain, Sweden, and Poland) with a higher sample size (ClinicalTrials.gov Identifier: NCT04592146) thus further assessment of the usability, UX and acceptance will be done, including extended work aimed at identifying ways of improving specific usability issues related to the individual answers to SUS, further exploring the relationship between usability and external factors (e.g., previous experience with technology, living conditions, etc.), and finding efficient ways to promote prompt responses to notifications. In addition, the home monitoring kit is in the process of being shifted towards a ubiquitous and transparent paradigm, that will probably maximize acceptance. These new devices will be based IoT technologies, easing their configuration, replacement, and scalability.

## Figures and Tables

**Figure 1 sensors-21-06458-f001:**
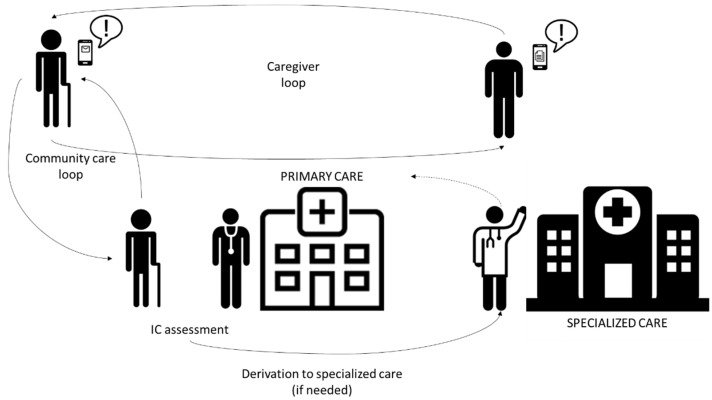
CAPACITY providers and interactions.

**Figure 2 sensors-21-06458-f002:**
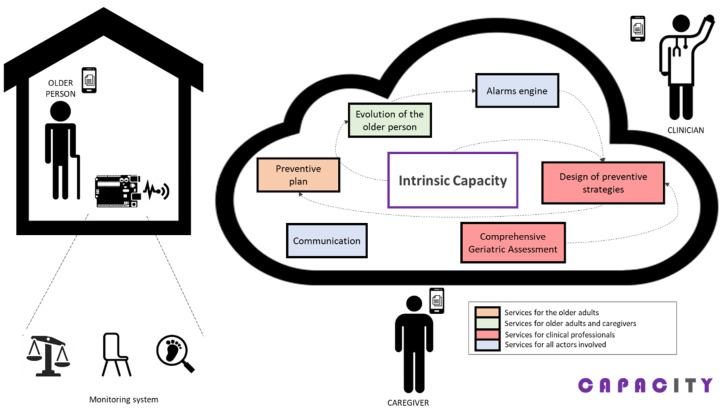
Conceptual architecture of CAPACITY.

**Figure 3 sensors-21-06458-f003:**
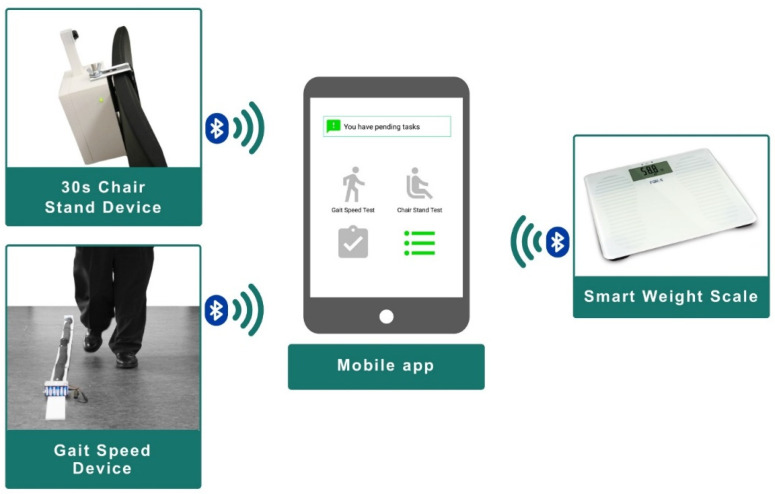
Home-monitoring kit.

**Figure 4 sensors-21-06458-f004:**
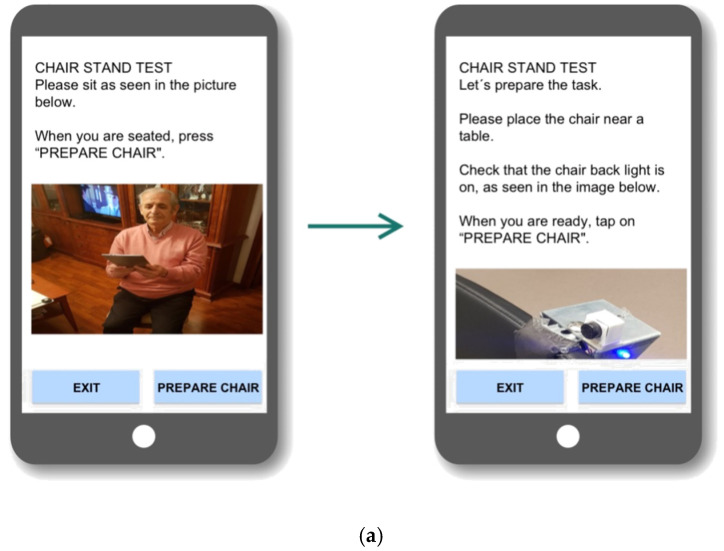
(**a**) First and (**b**) second—final—prototypes.

**Figure 5 sensors-21-06458-f005:**
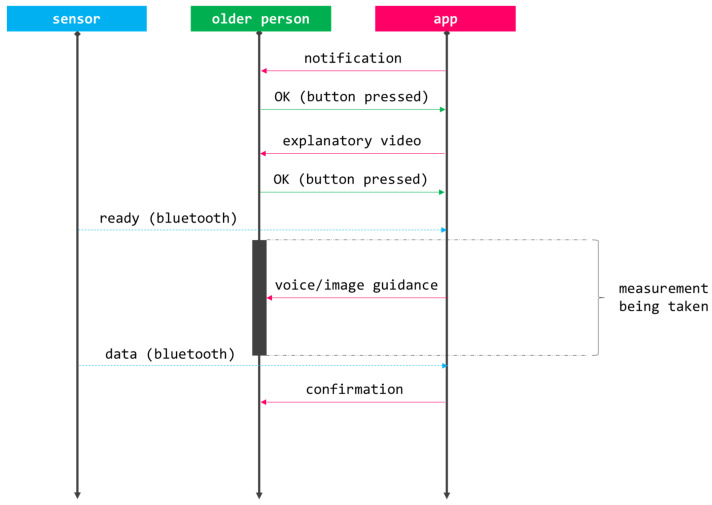
CAPACITY’s workflow to collect data from the home monitoring kit.

**Figure 6 sensors-21-06458-f006:**
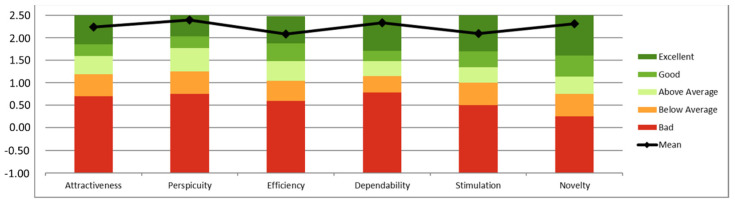
CAPACITY benchmarking according to UEQ.

**Table 1 sensors-21-06458-t001:** Description of the older population that participated in the study.

		*N*	Proportion (%)
**Sex**	Female	32	69.57
	Male	14	30.43
**Educational level**	No formal education	20	43.48
	Secondary education	20	43.48
	Primary education	5	10.87
	Higher education	1	2.17
**Experience with technology**	No experience	30	65.22
	Once or twice	4	8.70
	Occasional use	3	6.52
	Daily experience	9	19.56

**Table 2 sensors-21-06458-t002:** Average adherence to monitoring plan.

	Chair Stand	Weight Scale	Gait Speed	Frailty Phenotype Criteria	Barthel Index	FAQ	FRAIL Scale	MNA
**Default measurement periodicity (weeks)**	2	2	2	4	4	4	4	8
**Average adherence (%)**	96.38	97.44	95.70	100	99.54	95.95	94.44	94.91
**SD**	0.12	0.07	0.08	0.00	0.03	0.08	0.23	0.19

**Table 3 sensors-21-06458-t003:** SUS results.

SUS Item	Baseline	Month 3	Month 6
**Q1**	4.32	4.43	4.32
**Q2**	2.00	1.65	1.57
**Q3**	4.48	4.46	4.49
**Q4**	2.88	2.43	1.92
**Q5**	4.76	4.38	4.24
**Q6**	1.48	1.65	1.65
**Q7**	4.04	3.95	4.00
**Q8**	1.16	1.19	1.16
**Q9**	4.84	4.84	4.76
**Q10**	2.40	2.11	2.35
**Mean**	80.11	83.31	83.68
**SD**	13.66	15.07	11.62
** *P* _baseline-m3_ **	0.57		
** *P* _baseline-m6_ **	0.49		
** *P* _m3-m6_ **	0.88		

**Table 4 sensors-21-06458-t004:** Categorized UEQ results.

UEQ Category	Mean	SD	Confidence	Conf. Interval (*p* = 0.05)	
Attractiveness	2.20	0.64	0.21	1.99	2.40
Perspicuity	2.30	0.73	0.23	2.06	2.53
Efficiency	1.99	0.75	0.24	1.75	2.23
Dependability	2.16	0.66	0.21	1.95	2.38
Stimulation	2.05	0.72	0.23	1.82	2.28
Novelty	2.09	0.98	0.32	1.78	2.41

**Table 5 sensors-21-06458-t005:** TAM results.

	TAM item	Baseline	Month 3	Month 6
**Perceived usefulness**	Q1	4.16	4.35	4.16
	Q2	3.88	3.89	4.03
	Q3	3.84	4.16	4.24
	Q4	4.08	4.22	4.11
	Q5	4.12	4.27	4.30
	Q6	4.56	4.51	4.24
	Mean	24.64	25.41	25.08
	SD	4.79	3.68	4.48
	*P* _baseline-m3_	0.64		
	*P* _baseline-m6_	0.84		
	*P* _m3-m6_	0.40		
**Perceived** **ease-of-use**	Q1	3.72	4.30	4.24
	Q2	3.88	4.35	4.51
	Q3	4.36	4.30	4.65
	Q4	4.52	4.11	4.46
	Q5	3.72	4.00	4.16
	Q6	4.16	4.22	4.43
	Mean	24.36	25.27	26.46
	SD	4.86	5.16	3.77
	*P* _baseline-m3_	0.25		
	*P* _baseline-m6_	0.02		
	*P* _m3-m6_	0.36		
	Mean	49.00	50.68	51.54
	SD	8.24	6.68	6.97
	*P* _baseline-m3_	0.26		
	*P* _baseline-m6_	0.15		
	*P* _m3-m6_	0.83		

**Table 6 sensors-21-06458-t006:** Acceptance results (ad-hoc questionnaires).

Sensor		Q1	Q2	Q3	Q4	Q5
**Gait speed**	**Mean**	4.19	4.53	1.44	4.00	4.44
	**Std. dev**	0.75	0.61	0.84	0.99	0.88
**Chair stand**	**Mean**	4.61	4.64	1.08	4.53	4.69
	**Std. dev**	0.64	0.68	0.37	0.70	0.79
**Weight**	**Mean**	4.69	4.61	1.19	4.61	4.92
	**Std. dev**	0.58	0.90	0.82	0.69	0.28

**Table 7 sensors-21-06458-t007:** SUS and TAM evolution per category label.

	SUS at Baseline	SUS at M3	SUS at M6	TAM at Baseline	TAM at M3	TAM at M6
**Non-formal education**	75.75	80.16	82.33	46.55	48.56	50.40
**Primary education**	83.25	84.12	82.94	48.33	51.53	51.12
**Secondary education**	92.50	85.63	84.50	56.67	55.50	56.40
**Higher education**	80.00	-	-	53.00	-	-
** *p* **	0.31	0.69	0.94	0.24	0.14	0.24
**Living alone**	80.21	80.33	82.14	49.08	49.47	49.07
**Living with younger relatives**	90.00	88.75	90.63	50.50	54.50	54.75
**Living with other older person**	80.91	83.06	81.84	48.64	50.83	52.68
** *p* **	0.65	0.60	0.37	0.96	0.41	0.21
**Help from nobody**	84.04	87.61	84.89	48.77	52.05	51.50
**Help from a younger relative**	78.21	75.31	80.71	48.29	47.13	50.43
**Help from other older person**	78.13	77.50	79.64	50.75	50.50	53.29
**Help from social services**	80.00	60.00	77.50	50.00	50.00	48.00
** *p* **	0.79	0.052	0.66	0.97	0.37	0.84
**No previous experience with technology**	79.17	82.60	80.30	49.94	50.50	50.88
**Used technology once or twice**	92.50	85.83	94.17	52.00	52.67	56.33
**Occasional use of technology**	85.63	84.38	90.71	46.75	51.13	52.14
**Daily use of technology**	80.00	70.00	71.25	35.00	48.00	50.50
** *p* **	0.54	0.66	0.017	0.31	0.96	0.65
**Pre-frail**	80.00	84.50	83.89	49.69	51.40	53.00
**Frail**	82.71	82.27	82.59	48.25	50.56	51.07
** *p* **	0.62	0.66	0.57	0.67	0.86	0.87

## Data Availability

No new data were created or analyzed in this study. Data sharing is not applicable to this article.

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
