# Peer review of "Usability, User Experience, and Acceptance Evaluation of CAPACITY: A Technological Ecosystem for Remote Follow-Up of Frailty"

_sensors, 2021, doi:10.3390/s21196458_

Round 1

Reviewer 1 Report

Current work preconizes a  technological ecosystem called CAPACITY,  the CAPACITY has a very high-performance in term of usability, UX and acceptance. Results have been obtained in a real-world scenario, where frail older persons used CAPACITY as the main  vehicle to avoid transiting to disability.

Here some comments for a better version :

  1. It is required the foundation of a technological ecosystem. A conceptual model specifying the main proposal is required, given convincing and detailed arguments. It is required to raise the quality of presentation and research design for a better impact on the main proposal
  2. Frail older people as final user is important, but it is necessary to consider the medicine experience as an actor in the technological ecosystem. The authors should propose a better analyze, so that readers can understand what the differences / improvementsof the authors' work
  3. In order to appreciate more even current approach, compare results versus related work
  4. Restructure the conclusion indicating at least the main contribution, which objective (s) were reached, argue its impact as well as future work

Author Response

provided with track changes on and all reviews activated.

1

  • Information related to the CAPACITY technological ecosystem has been extended (please see paragraph starting in line 141), including a figure (Figure 2) depicting its conceptual architecture (A further description of the system can be found in reference [44]. This description has not been included in the current version to avoid duplicities):

[…]

Apart from older persons, CAPACITY also offers different functionalities to the other relevant stakeholders, namely primary and specialized care professionals, and informal caregivers, as shown in Figure 2.Work published in [44] contains a wider description of all functionalities an services offered by the CAPACITY technological ecosystem to all involved actors. In any case, this work solely focuses on the older persons as they are the center of the care.

[…]

2

  • We have updated the manuscript to clarify that the CAPACITY technological ecosystem supports an organizational model interconnecting all relevant actors in the care process: the older person, the informal caregiver, and the primary and specialized care professionals.  We believe that the modifications made to cover your requests in point 1 also clarify this point, since it is explicated both in the text and in the new figure (Figure 2), that apart from older persons, there are other users in the loop. We have also clarified that this work solely focuses on older persons since they are the center of the care (please see line 145):

[…] In any case, this work solely focuses on the older persons as they are the center of the care.

[…]

3

  • Discussion section has been updated relating our work with similar approaches (please see paragraph starting in line 457):

[…]

Not many RCTs exist in the field of ICTs applied to frailty management, which prevents the availability of a wider number of works focused on assessing the usability of technologies for treating frail population [42]. Works analyzing usability-related aspects of technology during real interventions also report satisfactory results [61,62], however, since the design procedure followed is scarcely described, the sample size is significantly smaller than the one presented in this paper, and data related to adherence are not optimal (ie. far from 100% adherence as reported in this paper), those results should be interpreted with caution. On the other hand, the majority of the research addressing how older persons interact with technology is done in controlled environments and under the supervision of domain experts [63–67]. Most of the published related research use standardized tools such as those used in this work, which is aligned with the methods followed in the current approach; moreover, despite the heterogeneous approaches in terms of the target application, ranging from rehabilitation [61–63] to exergames [64], monitoring cognitive impairment [65], fall risk [66] or evaluating available health apps [67], the used interaction instruments, including mobile devices [61,63,66,67], personal computers [62,65] or custom prototypes [64], and the diverse characteristics of the target population, that in some cases have a previous experience with the technology to be used [62] and in some others cannot use it without help [65], the UCD approach is a commonality backing almost all approaches from a methodological perspective.

4

  • Conclusions have been updated. It has been explicated the reached objective as well as the main contribution of our research work (please see paragraphs starting in line 477 and 487):

The objective of this research work was to investigate the usability, UX and acceptance of CAPACITY, a technological ecosystem to prevent disability. This objective has been achieved, obtaining very satisfactory results in all domains under study. The usability of CAPACITY (in terms of SUS) was rated as almost excellent, and UX (in terms of UEQ) as excellent; finally, the proposed technology, both from a software and a hardware perspective, seems to be highly accepted by the target population (in terms of TAM and ah-hoc questionnaires). Besides, adherence to using CAPACITY has been found optimal, which implies both that these superb results are correlated with maximizing the actual use of the proposed solution in a real environment and that data supporting the conclusions are based on reliable and solid information.

The main contribution of this paper is thus the demonstration that following an iterative UCD approach starting in a controlled laboratory environment to come up with a pre-validated interaction system, and later upscale it to a real uncontrolled environment is a valid strategy to maximize usability, UX, acceptance and actual adoption of a system. Furthermore, this research work contributes with a new experience to the scant number of RCTs studying how pre-frail and frail older persons interact with technology.

[…]

  • Modifications have been included to clarify future work (see from line 520 on):

[…] Future work includes carrying out a new multicentric field experimentation (RCTs in Spain, Sweden, and Poland) with a higher sample size (ClinicalTrials.gov Identifier: NCT04592146) thus further assessment of the usability, UX and acceptance will be done, including extended work aimed at identifying ways of improving specific usability issues related to the individual answers to SUS, further exploring the relationship between usability and external factors (eg. previous experience with technology, living conditions, etc.), and finding efficient ways to promote prompt responses to notifications. In addition, the home monitoring kit is in the process of being shifted towards a ubiquitous and transparent paradigm, that will probably maximize acceptance. These new devices will be based IoT technologies, easing their configuration, replacement, and scalability.

Reviewer 2 Report

The manuscript discusses a usability evaluation case study regarding the CAPACITY mobile application. Overall, the presented work is solid and methodologically rigid, providing a thorough analysis of the results. The participants' sample as well as the demographic distribution was appropriate, although the number of participants can be considered quite limited as the authors admit. However, the few findings of the evaluation process such as the issue mentioned with the ignored notifications would be very useful  for future similar works. Strangely these findings are not summarised in the conclusion section, in contrary to the limitations description which of course is very useful and appropriate to exist in this section, giving me the filling that the authors were slightly unfair to themselves. So, for the newer version of the manuscript authors are strongly recommended to further elaborate on the findings paragraph in the conclusion section. Finally, there is a typo in line 62, in particular the phrase "al risk of" should be corrected to "at risk of".

Author Response

Thank you very much for your feedback. It really helped improving the quality of our work.We have updated our manuscript accordingly for a better balance between findings/contributions and limitations. Please bear in mind line numbers are provided with track changes on and all reviews activated.

  • The following text has been added at the end of the discussion section to relate our work with similar experiences, supporting our approach (please see paragraph starting in line 457):

[…]

Not many RCTs exist in the field of ICTs applied to frailty management, which prevents the availability of a wider number of works focused on assessing the usability of technologies for treating frail population [42]. Works analyzing usability-related aspects of technology during real interventions also report satisfactory results [61,62], however, since the design procedure followed is scarcely described, the sample size is significantly smaller than the one presented in this paper, and data related to adherence are not optimal (ie. far from 100% adherence as reported in this paper), those results should be interpreted with caution. On the other hand, the majority of the research addressing how older persons interact with technology is done in controlled environments and under the supervision of domain experts [63–67]. Most of the published related research use standardized tools such as those used in this work, which is aligned with the methods followed in the current approach; moreover, despite the heterogeneous approaches in terms of the target application, ranging from rehabilitation [61–63] to exergames [64], monitoring cognitive impairment [65], fall risk [66] or evaluating available health apps [67], the used interaction instruments, including mobile devices [61,63,66,67], personal computers [62,65] or custom prototypes [64], and the diverse characteristics of the target population, that in some cases have a previous experience with the technology to be used [62] and in some others cannot use it without help [65], the UCD approach is a commonality backing almost all approaches from a methodological perspective.

  • We have also added the following text to the beginning of the conclusions section (paragraphs starting in line 477 and 487) to clarify what objective we have achieved as well as the contributions of the presented work:

The objective of this research work was to investigate the usability, UX and acceptance of CAPACITY, a technological ecosystem to prevent disability. This objective has been achieved, obtaining very satisfactory results in all domains under study. The usability of CAPACITY (in terms of SUS) was rated as almost excellent, and UX (in terms of UEQ) as excellent; finally, the proposed technology, both from a software and a hardware perspective, seems to be highly accepted by the target population (in terms of TAM and ah-hoc questionnaires). Besides, adherence to using CAPACITY has been found optimal, which implies both that these superb results are correlated with maximizing the actual use of the proposed solution in a real environment and that data supporting the conclusions are based on reliable and solid information.

The main contribution of this paper is thus the demonstration that following an iterative UCD approach starting in a controlled laboratory environment to come up with a pre-validated interaction system, and later upscale it to a real uncontrolled environment is a valid strategy to maximize usability, UX, acceptance and actual adoption of a system. Furthermore, this research work contributes with a new experience to the scant number of RCTs studying how pre-frail and frail older persons interact with technology.

[…]

  • Further work related to the notifications is included in the last paragraph of the manuscript (please see line 527):

[…] finding efficient ways to promote prompt responses to notifications […].

Round 2

Reviewer 1 Report

Thank you for your pertinent answers to my comments!

Author Response

Thank you for your suggestions. In this case, we would like to clarify that self-citations have not been included to enlarge the number of cites received by the authors but because most of the previous work of Prof. Rodríguez-Mañas' research group sets the clinical grounds of our topic of interest. However, we realized a couple of those references either were covered by others or could be removed since the idea being backed is not crucial for this paper. Please find below the reasons behind keeping or removing the references (bear in mind numbers refer to previous version of the manuscript).

[3] - This is a very relevant clinical publication in the domain of frailty given the collaboration of Prof. Rodríguez-Mañas and Prof. Fried, the highest world authority in frailty. This publication perfectly frames frailty so referencing it is mandatory.

[4] - The joint action ADVANTAJE has provided a common approach model of care to face the challenge of frailty of older people within a common European framework. The fresher data on prevalence of frailty have been published by them.

[6-7] - These are publications which the first author is Prof. Izquierdo, one of the most relevant figures in the world in the domain of physical activity. The VIVIFRAIL physical activity program, that is used as one of the intervention pillars of our system is described. Results obtained by this project (success story of the EU) support the arguments used in this paper, so its citation is justified.

[17] - Certainly the sentence where that uses this reference is not relevant for this paper. We have removed it.

[18] - Few studies tackle frailty from a neurological perspective and this work does. Our research group has collaborated with many others in this domain, creating synergies that have lead to discover new relevant knowledge.

[44] - It is not the aim of this paper fully describing the CAPACITY interaction system but the usability, acceptance and UX results obtained after using it in a real-world scenario. We consider this reference relevant to forward the reader to the details of the functionalities and services offered by the technological ecosystem under use.

[45] - In this work we evaluate the usability, UX and acceptance not only of a software but also of the hardware that older persons use to undergo the program. We understand this reference to forward the reader to more details on the gait speed sensor in terms of design and validation.

[46] - This reference has been removed since [47] covers the contents.

[47] - The same as in the gait speed sensor, we understand this reference is necessary to forward the reader to more details on the chair stand sensor (to measure power in the lower limbs) in terms of design and validation.

[48] - One of the main conclusions of this paper is that the followed UCD process has maximized the usability, UX and acceptance. This paper describes the iterative method we have followed to reach the version of the system that has been validated, so we consider it relevant to be cited. By mistake, this reference appeared twice in the paper. Now it only appears once where pertinent.